# Lung Tumor Location and Identification with AlexNet and a Custom CNN

## Abstract

Lung cancer is the leading cause of cancer deaths in the world and early detection is a crucial part of increasing patient survival. Deep learning techniques provide us with a method of automated analysis of patient scans. In this work, we compare AlexNet, a multi-layered and highly flexible architecture, with a custom CNN to determine if lung nodules with patient scans are benign or cancerous. We have found our CNN architecture to be highly accurate (99.79%) and fast while maintaining low False Positive and False Negative rates ($< 0.01\%$ and $0.15\%$ respectively). This is important as high false positive rates are a serious issue with lung cancer diagnosis. We have found that AlexNet is not well suited to the problem of nodule identification, though it is a good baseline comparison because of its flexibility.

## 1 Introduction

Lung cancer is the leading cause of cancer deaths in the world with approximately 1.69 million deaths due to lung cancer each year( who (2017)). Early detection is a crucial part of increasing the chance of patient survival and automated processing techniques are an essential part of this process. Detecting and identifying lung nodules as benign or cancerous is a difficult task. Due to the nature of the current screening methods, normal tissues or non-cancerous masses can be misidentified as can malignancies. Our goal is to accurately locate and identify benign and cancerous growths in patient images. High false positive rates are also a very serious issue since false positive rates in lung cancer diagnoses are as high as 95% clinically( Rivera et al. (2013); Moyer (2014)). The high rates of false positive diagnoses can lead to unnecessary treatments and procedures which wastes resources and cause undue stress to patients and families.

AlexNet is one of the most famous of the CNN architectures( Krizhevsky et al. (2012)). It has a multi-layered architecture that consists of five convolutional layers, with three max pooling layers, and three fully connected layers which allow it to identify up to 1000 different objects from within images. This network has quickly become a standard since it's award winning introduction in 2012. The network which has been trained over millions of images, is highly flexible and can be applied to many different classification problems.

Applying this to to task of cancer nodule classification, however, is somewhat more difficult. The generalized nature of AlexNet means it's not best suited to such specialized problem. Here we compare our CNN architecture with the performance of AlexNet in classifying lung nodules as cancerous or benign.

## 2 Methods

### 2.1 Dataset

The Luna 2016 Challenge dataset was derived from the LIDC-IDRI dataset( lid (2015); Setio et al. (2017)). The dataset includes 888 CT scans. Benign and cancerous tumors were annotated with each annotation being confirmed by at least three radiologists. The data is available in both MDH and DICOM formats. We have used this for our training and testing sets as it is well annotated. Figure 1 shows examples of cancerous and benign growths.

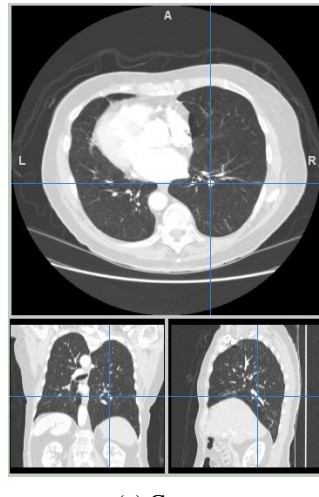
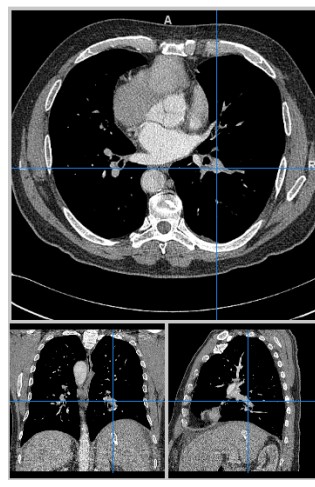

(a) Cancer                                                   (b) Benign

Figure 1: These images show examples of both cancerous and benign growths. (a) Shows cancerous tumors and (b) shows benign growths. The intersection of the blue lines shows the center of an example nodule in each image.

We preprocessed our images by converting them from their MDH and RAW formats to a PNG format. This allows us to take advantage of Matlab's image data storage object( Mathworks (2016)). This allows Matlab to process certain image types(JPG and PNG) for the CNN without having to import them all into the memory at the same time. The image data store object enables us to more efficiently build our networks with large datasets. We have chosen the PNG format because of the lossless nature of the format. No other pre-processing or modifications were done to the images in this step.

We also needed to convert the annotations of the dataset from the patient coordinates to the image coordinates in order to have the locations of the tumors in the PNG images. The images in this dataset are free of rotations therefore this conversion is simple. We can calculate these coordinates using the following equation:

$$C_I = \frac{C_w - origin}{spacing}$$

where $C_I$ are the image coordinates, $C_w$ are the world coordinates, origin is the image origin, and spacing is the pixel spacing. We can retrieve the slice size information from the DICOM images and use this and the converted coordinate system to determine the location of the tumors in the PNG images.

## 2.2 CNN ARCHITECTURE

The architecture of the the CNN we have designed for nodule classification is shown in Figure 2.

Here we explain each step of our CNN architecture, as seen in Figure 2, in detail.

1. The medical scans and the labels for the images (cancerous or benign) are imported into the system for training and testing.

2. Randomized testing and training sets are created for the CNNs. We use a data split of 70% for training (which is approximately 622 sets of scans). The rest of the data was used to create a test set of 266 scans. All training was done using four Nvidia GTX 1080 GPUs.

3. This convolution layer uses 64 10X10 convolutions with a 1X1 stride and 5X5 padding to further convolve the features folowed by a Rectified Linear Unit (ReLU) layer to set all negative elements to zero.

4. The convolved features then go into the maximum pooling layer. The pooling layer calculates the maximum value of the feature over a region of the image so we can use the features for classification. This max pooling layer has a filter size of 3X3.

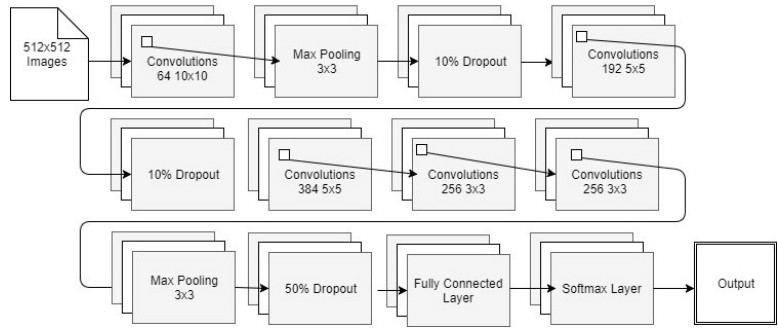

Figure 2: This image shows the architecture of the CNN for nodule detection.

5. The next layer is a 10% dropout layer to reduce the over-fitting.

6. The next convolution layer uses 192 5X5 convolutions with a 1X1 stride and 5X5 padding to further convolve the features. This is followed by a ReLU layer.

7. The convolved features then go into another maximum pooling layer, this one of size 2X2.

8. The next layer is a 10% dropout layer to reduce the over-fitting.

9. Convolution 384 5X5 convolutions with stride 1X1 and no padding. This is followed by a ReLU layer.

10. The next convolution layer uses 256 3X3 convolutions with a 1X1 stride and no padding to further convolve the features. This is followed by a ReLU layer.

11. The next convolution layer uses 256 3X3 convolutions with a 1X1 stride and no padding to further convolve the features. This is followed by a ReLU layer.

12. The next convolution layer uses 256 3X3 convolutions with a 1X1 stride and no padding to further convolve the features. This is followed by a ReLU layer.

13. The next convolution layer uses 128 3X3 convolutions with a 1X1 stride and no padding to further convolve the features. This is followed by a ReLU layer.

14. The convolved features then go into the last maximum pooling layer with a filter size of 3X3.

15. The next layer is a 50% dropout layer.

16. The features resulting from the convolution and pooling of the features of each CNN are then input into their respective softmax layers. The softmax layer calculates the normalized exponential function to calculate the output activation function. This function is calculated as:

$$P(c_r|x,\theta) = \frac{P(x,\theta|c_r)p(c_r)}{\sum_{j=1}^{k} P(x,\theta|c_j)p(c_j)} = \frac{exp(a_r(x,\theta)}{\sum_{j=1}^{k} exp(a_j(x,\theta)}$$

Where $P(c_r)$ is the class prior probability, $P(x,\theta|c_r)$ is the conditional probability of the given class $r$, and $a_r = ln(P(x,\theta|c_r)P(c_r))$( Bishop (2006)).

17. The final layer of the CNNs is the prediction layer that outputs the predicted value for each image.

18. The CNN is then fed into the RCNN object detector to locate and identify both the benign and cancerous nodules.

The series of 3X3 convolutional layers work as segmentation of the images. A CNN based segmentation method based on Convolutional Neural Networks(CNN) was adapted from deep CNN method for brain image segmentation( Pereira et al. (2016); Simonyan & Zisserman (2014)). This design uses multiple layers of 3X3 filters to give a deeper architecture and reduces over-fitting. We have found that using several of these convolutional layers decreased our False Positive Rate significantly.

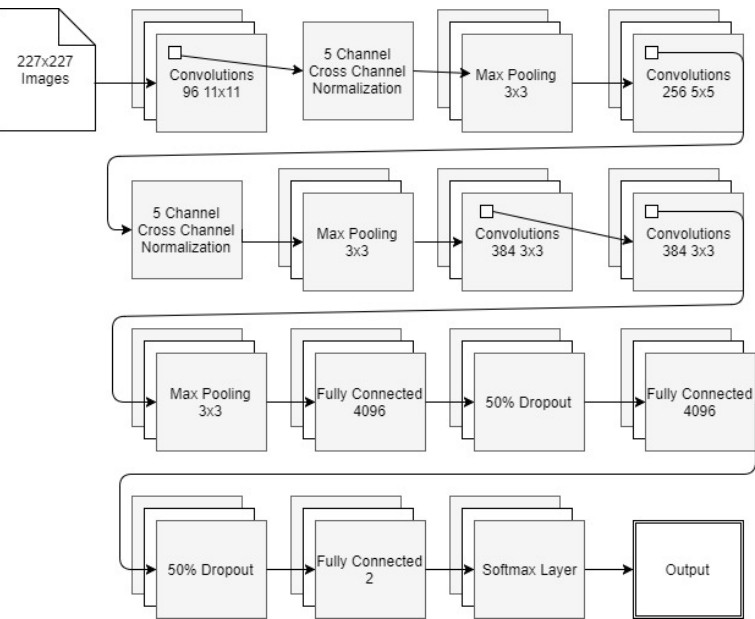

Figure 3: This image shows the architecture of AlexNet.

### 2.3 ALEXNET

AlexNet is one of the most famous of the CNN architectures( Krizhevsky et al. (2012)). It has a multi-layered architecture that consists of five convolutional layers, with three max pooling layers, and three fully connected layers. This highly flexible network can identify 1000 different objects within images. The network can also have certain layers retrained in order to identify objects within images not from the original training set. Figure 3 shows the architecture of AlexNet.

### 2.4 ACCURACY EVALUATION

We use several different metrics in order to evaluate the accuracy of our model. We have calculated the accuracy, precision, and recall( Powers (2011)). The accuracy gives the overall percentage of how many correct prediction our model outputs. Precision is a measurement of true versus false positives while recall is a measurement of the positive predictions to the actual positives. We also use the following abbreviations in the rest of this paper: $tp$ is the number of true positives, $tn$ is the number of true negatives, $fp$ is the number of false positives, and $fn$ is the number of false negatives.

Accuracy alone isn't a enough to fully assess the strength of a classifier. Therefore, we use the $F_1$ score and the Matthews Correlation Coefficient (MCC) to determine how well each classifier is doing with our dataset. The $F_1$ score is a quality measure that combines the precision and recall measures into one value. The MCC gives us a measure of the quality of our binary classifier.

The accuracy of the models is calculated using the formula:

$$accuracy = \frac{tp + tn}{tp + tn + fp + fn}$$

We calculate the accuracy of the individual CNNs, the unsmoothed and the smoothed, in addition to the overall accuracy after voting.

The precision is the ratio of true positives and the total positive predictions.

$$Precision = \frac{tp}{(tp + fp)}$$

The recall is the ratio of true positives and the total, actual positives.

$$Recall = \frac{tp}{(tp + fn)}$$

The $F_1$ score( Powers (2011) is a combination of these two measures. It is calculated as:

$$F_1 = 2 * \frac{Recall * Precision}{Recall + Precision}$$

where recall and precision are calculated as above.

The $F_1$ score ranges from 0 to 1, where 1 is the best possible score.

The Matthews Correlation Coefficient (MCC) score( Powers (2011) is calculated as:

$$MCC = \frac{(tp/n) - sp}{\sqrt{ps(1 - s)(1 - p)}} = \frac{(tp)(tn) - (fp)(fn)}{\sqrt{(tp + fp)(tp + fn)(tn + fp)(tn + fn)}}$$

where $n$ is the total number of predictions, $s = (tp + fn)/n$, $p = (tp + fp)/n$, $tp$ is the number of true positives, $tn$ is the number of true negatives, $fp$ is the number of false positives, and $fn$ is the number of false negatives.

The MCC gives us an alternate measure of accuracy by providing a measure of the quality of the predictions: it return values between $-1$ and 1. An MCC value of 1 is a perfect prediction while 0 is no better than a random prediction. An MCC value between 0.4 and 0.6 is considered a strong classifier while an MCC value of over 0.7 is considered a very strong classifier.

The False Positive Rate (FPR)( Powers (2011) is calculated as:

$$FPR = fp/(tn + fp)$$

The False Negative Rate (FNR)( Powers (2011) is calculated as:

$$FPR = fn/(tp + fn)$$

## 3 RESULTS AND DISCUSSION

AlexNet is one of the gold standards of CNNs( Krizhevsky et al. (2012)). We use the network to classify the images as containing either benign or cancerous tumors. The retraining process for AlexNet takes approximately 19.61 hours with the Luna dataset. Table 1 shows the accuracy and quality metrics for the classification with the retrained AlexNet for each of the Luna subsets. We can see that the accuracy, as well as the precision and recall, are high for this classifier. The average accuracy is 99.69%.

Table 2 shows the accuracy and quality metrics for our custom CNN using the Luna Dataset subsets. It is very important to note that the training process for our CNN took 1.6 hours using the Luna dataset. This is significantly less time than the retraining process for AlexNet. We also see that our accuracy, recall, and precision are all very high. We have an average accuracy of 99.78%, slightly higher than that of AlexNet. The precision is also extremely high, indicating we have a very low false positive rate.

Table 3 shows the comparison between our simple CNN and the AlexNet. We can see that the accuracy, precision, and recall are comparable for the two networks: the major differences are in the training time and the False Positive Rates (FPR). The CNN we designed for the RCNN takes significantly less time than retraining the AlexNet for our images. It is also $13X$ faster to train our simple CNN than to retrain AlexNet. We also have from Table 1 and Table 2 that the classification time is less with our CNN design as well.

Arguably, the most important difference between the two networks is the False Positive Rate. High false positive rates are a serious issue with lung cancer diagnosis with rates as high as 95% being seen clinically( Rivera et al. (2013); Moyer (2014). The CNN we designed for this task has a significantly lower false positive rate than AlexNet. AlexNet's false positive rate of 54.26% is too high to be a good option for this use since we want our FPR to be as low as possible. It is also

Table 1: Accuracy and quality metrics for the AlexNet classifier with the Luna Data Subsets[a].

| Subset | Accuracy | Precision | Recall | Classification Time |
|---|---|---|---|---|
| 0 | 99.69% | 99.84% | 99.84% | 769.47s |
| 1 | 99.76% | 99.98% | 99.77% | 247.52s |
| 2 | 99.77% | 99.90% | 99.86% | 307.49s |
| 3 | 99.78% | 99.93% | 99.85% | 276.80s |
| 4 | 99.79% | 99.91% | 99.88% | 274.69s |
| 5 | 99.44% | 99.68% | 99.76% | 255.23s |
| 6 | 99.87% | 100.00% | 99.87% | 243.83s |
| 7 | 99.55% | 99.79% | 99.76% | 242.40s |
| 8 | 99.60% | 99.82% | 99.77% | 208.89s |
| 9 | 99.62% | 99.83% | 99.79% | 241.33s |

[a] $Accuracy = (tp + tn)/(tp + tn + fp + fn)$, $Precision = tp/(tp + fp)$, and $Recall = tp/(tp + fn)$. Runtime is the time recorded by the GPU at run time in seconds.

Table 2: Accuracy and quality metrics for the CNN classifier[b].

| Subset | Accuracy | Precision | Recall | Classification Time |
|---|---|---|---|---|
| 0 | 99.80% | 100.00% | 99.80% | 537.21s |
| 1 | 99.76% | 100.00% | 99.76% | 90.47s |
| 2 | 99.83% | 100.00% | 99.83% | 104.29s |
| 3 | 99.81% | 100.00% | 99.81% | 99.81s |
| 4 | 99.85% | 100.00% | 99.85% | 99.27s |
| 5 | 99.70% | 100.00% | 99.70% | 93.62s |
| 6 | 99.86% | 100.00% | 99.86% | 94.66s |
| 7 | 99.73% | 100.00% | 99.73% | 91.99s |
| 8 | 99.76% | 100.00% | 99.76% | 82.36s |
| 9 | 99.74% | 100.00% | 99.74% | 93.60s |

[b] $Accuracy = (tp + tn)/(tp + tn + fp + fn)$, $Precision = tp/(tp + fp)$, and $Recall = tp/(tp + fn)$. Runtime is the time recorded by the GPU at run time in seconds.

Table 3: Comparison of AlexNet and the custom CNN performance[c].

| Network | Accuracy | Precision | Recall | FPR | FNR | Training Time |
|---|---|---|---|---|---|---|
| AlexNet | 99.72% | 99.93% | 99.85% | 54.26% | 0.15% | 19.60h |
| Custom CNN | 99.79% | 100.0% | 99.85% | 0.00% | 0.15% | 1.61h |

[c] $Accuracy = (tp + tn)/(tp + tn + fp + fn)$, $Precision = tp/(tp + fp)$, $Recall = tp/(tp + fn)$, $FPR = fp/(tn + fp)$, and $FPR = fn/(tp + fn)$. Runtime is the time recorded by the GPU at run time.

Table 4: Comparison of AlexNet and the custom CNN classification strength[d].

| Network | $F_1$ | MCC |
|---|---|---|
| AlexNet | 0.99 | 0.37 |
| Custom CNN | 0.99 | 0.56 |

[d] $F_1$ has a range from 0 to 1, scores above 0.6 are considered strong classifiers. Matthews Correlation Coefficient has a range from $-1$ to 1, where scores above 0.4 are considered strong classifiers.

important to note that our CNN also has a stronger Matthews Correlation Coefficient, as we see in Table 4. This indicates that while AlexNet is doing much better than random guessing, it is not a strong classifier for this dataset. This classification strength measure combined with the FPR rates indicates our classifier is a better option than AlexNet in this case.

## 4 CONCLUSION

AlexNet marks a serious advancement in image identification. Though as we see from our results here, AlexNet does better with more generalized image identification. The high false positive is particularly indicative of this issue even though the accuracy is good. The strength statistics of the classifier also shows that our custom CNN for this problem is a much stronger classifier. AlexNet's strength lies in it's ability to identify diverse and wide ranging objects in images but it stumbles when faced with less diversity and a more subtle classification problem. We were able to develop a strong classifier that is both fast and accurate while maintaining low False Positive and False Negative Rates.

The next step is to use the CNN we developed to train a Regions with CNN features(RCNN) network to automatically locate the tumors within scans. We have obtained the bounding boxes of the nodules within our scans using the coordinate and annotation information. This allows us to save considerable time as manually identifying the nodules, even with the coordinate information, is tedious and difficult work. We also plan on further developing this technique to classify cancerous nodules by subtype.

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
