# OpenReview forum: "Lung Tumor Location and Identification with AlexNet and a Custom CNN"
_ICLR.cc/2018/Conference — Reject_

### Official Review · AnonReviewer1 · 2017-11-22
**a trivial comparison of 2 CNN models for lung cancer detection on CT scans**

**Rating:** 2
**Confidence:** 5

**Review:**

This paper compares 2 CNN architectures (Alexnet and a VGG variant) for the task of classifying images of lung cancer from CT scans. The comparison is trivial and does not go in depth to explain why one architecture works better than the other. Also, no effort is made to explain the data beyond some superficial description. No example of input data is given (what does an actual input look like). The authors mention "the RCNN object detector" in step 18, that presumably does post-processing after the CNN. But there is no explanation of that module anywhere. Instead the authors spend most of the paper listing in wordy details the architecture of their VGG variant. Also, a full page is devoted to detailed explanation of what precision-recall and Matthews Correlation Coefficient is! Overall, the paper does not provide any insight beyond: i tried this, i tried that and this works better than that; a strong reject.

---

### Official Review · AnonReviewer2 · 2017-11-26
**Paper with interesting ideas, but far from meeting ICLR level.**

**Rating:** 3
**Confidence:** 4

**Review:**

The authors compare a standard DL machine (AlexNet) with a custom CNN-based solution in the well known tasks of classifying lung tumours into benign or cancerous in the Luna CT scan dataset, concluding that the proposed novel solution performs better.
The paper is interesting, but it has a number of issues that prevents it from being accepted for the ICLR conference.

First, the scope of the paper, in its present form, is very limited: the idea of comparing the novel solution just with AlexNet is not adding much to the present landscape of methods to tackle this problem.
Moreover, although the task is very well known and in the last few year gave rise to a steady flow of solutions and was also the topic of a famous Kaggle competition, no discussion about that can be found in the manuscript.
The novel solution is very briefly sketched, and some of the tricks in its architecture are not properly justified: moreover, the performance improvement w.r.t . to AlexNet is hardly supporting the claim.
Experimental setup consists of just a single training/test split, thus no confidence intervals on the results can be defined to show the stability of the solution.
The whole sections 2.3 and 2.4 include only standard material unnecessary to mention given the target venue, and the references are limited and incomplete.
This given, I rate this manuscript as not suitable for ICLR 2018.

---

### Official Review · AnonReviewer3 · 2017-11-27
**difficult to read, need more details**

**Rating:** 3
**Confidence:** 4

**Review:**

The paper compares AlexNet and a custom CNN in predicting malignant lung nodules, and shows that the proposed CNN achieves significantly lower false positives and false negative rates.

Major comments

- I did not fully understand the motivation of the custom CNN over AlexNet.

- Some more description of the dataset will be helpful. Do the 888 scans belong to different patients, or same patient can be scanned at different times? What is the dimensionality of each CT scan?

- Are the authors predicting the location of the malignant nodule, or are they classifying if the image has a malignant nodule? How do the authors compute a true positive? What threshold is used?

- What is 'Luna subsets'? What is 'unsmoothed and smoothed image'?

Minor comments

- The paper is difficult to read, and contains a lot of spelling and grammatical errors.

---

### Public Comment · (anonymous) · 2017-12-01
**Reproducibility Challenge**

Hello,

I would like to reproduce your results as part of the ICLR 2018 Reproducibility Challenge (http://www.cs.mcgill.ca/~jpineau/ICLR2018-ReproducibilityChallenge.html).

Would you be able to share your Code with me? Or any other helpful material for that matter?
You mentioned some pre-processing of the data (converting to PNG and annotating image with coordinates), would you be able to share the scripts that do so?

Can you give me some information about the computational resources needed to run the analysis (CPU/GPU, RAM,...)?

Thank you very much

---

### Public Comment · (anonymous) · 2017-12-15
**Reproducibility Challenge**

We have attempted to reproduce the following manuscript entitled Lung Tumor Location and Identification with AlexNet
and a Custom CNN, as part of the Reproducibility Challenge sponsored by our Machine Learning course. We find the paper
interesting in that it focuses on automating Cancer diagnosis from Lung CT scans which would perhaps have real clinical impact on the lives of patients.

The paper was difficult to reproduce due to missing key information. Code and input data were not included with the
manuscript. The software packages used were not always mentioned and software versions were not shared. Computational
resources needed to run the analysis were also not disclosed. Hyper-parameters of the CNN models were sometimes missing, such as the Random Seed used throughout the analysis, the number of trained epochs for each CNN and the optimizer used to compile the models.

The authors mention that they split the dataset into 70% training data and 30% test data, however, they did so based on
CT scans and not based on instances of the labels. This was not described in enough detail to allow proper reproduction. Some
more details would have been useful.

The authors described their CNN structure in great detail; however, there was a mismatch between the description in the
text and the figure representing the CNN architecture. Two convolutional layers, steps 12 and 13, were not displayed in the figure.

The authors mention that they also attempted to localize the tumor across the CT scan and mention using a RCNN,
however, no other details were given and hence this part was difficult to reproduce.

The authors used an input size of 512 x 512 for their CNN but used 227 x 227 input size for AlexNet. It was not
described how the transformation was done or whether input data was processed separately at two different sizes for each CNN.

Almost 80% of the time spent on this challenge was spent on understanding and processing the input data, in order to
make it ready to train the CNN. Input data is a fundamental part of Machine Learning and small differences in data processing
might result in different datasets and thus different performance by the algorithm. Given the complexity of the dataset and the difficulty in describing all steps in details, it is advisable that the processed data used to train a model be shared in its exact format or that the code that produced the data be made available to avoid small discrepancies.

The authors have reported an accuracy of about 99.79%, roughly 3% higher than anything reported online to date,
including the scoreboards of the Grand LUNA Challenge and the Kaggle Data Science Bowl 2017. It would be interesting for the authors to submit their results to both of these scoreboards to register their performance.



Note: We are researchers ourselves and have authored manuscripts which if judged today, would likely have missing
information that would make them hard to reproduce as well. This has been an interesting mental exercise as your readers, but also for us as authors of scientific content. While it might not be fun to go through this task of thinking of reproducibility and what details to include in a manuscript, it certainly is necessary for all of us as researchers to ensure that our work has the highest impact on the scientific community and society as a whole. I hope you appreciate the value of what we are trying to do and not take any comments personally; this is really more of an issue of us as a community and a need to standardize scientific reporting and instigate thought about reproducibility.

---

### Public Comment · (anonymous) · 2017-12-16
**Reproducibility Challenge**

The motivation for the paper was a challenge to improve on the classification of CT scans whether they contain cancerous
or benign tumors. The publicly available and well-labeled dataset that the paper uses is the modified LIDC-IDRI dataset that
contains sets of CT scans of patients with lung tumours. The size of the dataset is 128GB of DICOM images, and is 32GB
when converted to PNG. Manipulation of this data was difficult and time consuming on its own.
The authors obtained their input and output sets from the closed Luna 2016 Challenge, which came in a preprocessed format
and also came divided into 10 different subsets. We did not have access to the Luna 2016 Challenge dataset (no public access),
so we ended up using the publicly available LIDC-IDRI dataset and pre-processed it in a way that matched the Luna dataset
description. The authors state that they worked with 888 patients, whereas for our dataset, following the same procedure,
yielded 896 patients. This discrepancy in input was not so big that we would expect significantly different results. The only
implication is that some additional patient files were removed before finalizing the Luna dataset in a manner that was not
documented. Converting the data to PNG files was needed as the original format (DICOM and MDH) contains more than just
the pictures of the scans, which we were able to achieve. We also had to extract the output labels from the public dataset
ourselves, which could have also contributed to some differences from the data that the paper uses. Given that the authors of
the paper had to convert the pixel labels from millimeters to pixels, we believe the labels provided to the authors differed in
format from those supplied with the LIDC-IDRI.
The paper implemented two CNN architectures: an AlexNet modified to binary output, and a custom CNN designed by
the authors for the purpose of tumour identification. The architecture was described using both figures and a layer-wise text
description, however these descriptions were inconsistent. We used a version that contained all the layers from both architectures,
which adds another uncertainty in the reproduction of the results. The motivation for using the custom CNN was the ability
to decrease the false positive rates compared to the baseline AlexNet. Ultimately, our findings were unable to support or deny
this assertion. In our efforts to trying to run both of these models, the difficulty that we faced (apart from the ambiguity in the
network architecture) was the lack of proper hardware available to us. We were able to use a NVIDIA Titan X GPU, which
only provided 12GB of memory that turned out to be a limiting factor for us as we needed to decrease the resolution of our
input images for the custom CNN. We would also like to note the lack of the hyper-parameters identified in the paper of which
were used to train the two classifiers.

---

### Public Comment · (anonymous) · 2017-12-16
**ICLR2018 Reproducibility Challenge**

Greetings to the authors,

McGill University’s COMP 551 Applied Machine Learning class has decided to conduct a reproducibility challenge, and our team has decided to analyze the paper "Lung Tumor Location and Identification with AlexNet and a Custom CNN".

Lung cancer is the leading cause of cancer deaths in the world, with roughly 1.67 million deaths a year. Recent development of machine learning methods in medical imaging have provided doctors with a complementary set of tools to tackle these issues.
However, existing CNN architectures may not be optimal for specialized medical tasks such as cancer nodule classification. Thus, the authors propose a new architecture and compare it to AlexNet, an existing architecture.

As such, we seek verify the claims by the original authors which state that the proposed CNN architecture classifies cancer nodes at an accuracy of 99.79 and that false positive rates are reduced to less than 0.01%.

The dataset utilized, as outlined by the original authors are derived from the LUNA2016 dataset, part of the LIDC/IDRI database. We were able to obtain the 888 scans, along with the annotations, and are fairly confident that this is the same data set the authors used. However, the authors did not provide sufficient information in order to perform the preprocessing of the raw images. Therefore, our team followed LUNA2016's tutorials which explain the procedure to convert relevant sections of the medical images into either PNG format. We assumed that the authors followed the same method as in the tutorial.

In order to reproduce the authors' results, we constructed a network following the proposed architecture from their paper. In this respect, the authors provided very clear and concise  instructions about the architecture of their CNN. Our CNN was implemented using Keras with TensorFlow backend instead of MATLAB, as we assume the authors used. Every layer of our network follows the description of the authors' network, with the exception of the fully-connected layers due to the lack of specifications. Thus, our architecture contains 7 convolutional layers, along with their associated padding, pooling and dropout layers. The authors did not specify the number of dense layers at the end of their convolutional layers, nor the activation functions for the dense layers. Thus, we opted for flattening the convolutional layer, and connecting it to two fully-connected layers. Due to the output being binary, a Sigmoid activation function with binary cross-entropy loss function. It is also worth noting that the figure provided in the original paper differs from the architecture presented. We followed the written architecture instead of the graphical one. The model was then trained on an NVidia GTX970 GPU, which is less powerful than the four NVidia GTX1080 which the authors used. The authors clearly stated the time requirements for training their networks.

The original paper provides an extensive validation table. However, the procedures were unclear as our team could not understand how exactly the authors obtained their validation results. Thus, we decided to work with the subsets contained in the data set. On the 10 total subsets, we performed 7-fold cross validation, using 6 subsets as training sets, 1 subset as validation set each time, and withholding 3 subsets for a final test set. Our results turned out similar to the ones in the paper. However, in addition, we propose a baseline consisting of predicting all tumors as benign. Such a baseline performs significantly better than a random baseline, and due to class imbalance in the data set, it provides a comparable accuracy to the CNN's.

As a final note, the authors' paper contains sections which are well-described and detailed such as the architecture of their proposed model. However, other sections may be lacking in clarity and could be ambiguous, notably in their description of validation procedures. Moreover, there are discrepancies between the architecture of the proposed CNN and the figure showing the graphical model. Minor typos and mistakes include the extension of the data set, which is ".MHD", not ".MDH", and the confusion of FPR and FNR in the explanations for the evalutation metrics.

Overall, the authors' approach to the classification of tumors was interesting, and we would have liked to reproduce the authors' results more faithfully if the code behind the results was provided.

Our full review is located at: https://github.com/ExTee/COMP551-Final

---

### Decision · Program_Chairs · 2018-01-29
**ICLR 2018 Conference Acceptance Decision**

**Decision:**

Reject

**Comment:**

Pros:
- Addresses an important medical imaging application
- Uses an open dataset

Con:
- Authors do not cite original article describing challenge from which they use their data: https://arxiv.org/pdf/1612.08012.pdf , or the website for the corresponding challenge: https://luna16.grand-challenge.org/results/
- Authors either 1) do not follow the evaluation protocol set forth by the challenge, making it impossible to compare to other methods published on this dataset, or 2) incorrectly describe their use of that public dataset.
- Compares only to AlexNet architecture, and not to any of the other multiple methods published on this dataset (see: https://arxiv.org/pdf/1612.08012.pdf).
- Too much space is spent explaining well-understood evaluation functions.
- As reviewers point out, no motivation for new architecture is given.